# New Hybrid Phenalenone Dimer, Highly Conjugated Dihydroxylated C_28_ Steroid and Azaphilone from the Culture Extract of a Marine Sponge-Associated Fungus, *Talaromyces pinophilus* KUFA 1767

**DOI:** 10.3390/md21030194

**Published:** 2023-03-21

**Authors:** Fátima P. Machado, Inês C. Rodrigues, Aikaterini Georgopolou, Luís Gales, José A. Pereira, Paulo M. Costa, Sharad Mistry, Salar Hafez Ghoran, Artur M. S. Silva, Tida Dethoup, Emília Sousa, Anake Kijjoa

**Affiliations:** 1ICBAS—Instituto de Ciências Biomédicas Abel Salazar, Rua de Jorge Viterbo Ferreira, 228, 4050-313 Porto, Portugal; 2Interdisciplinary Centre of Marine and Environmental Research (CIIMAR), Terminal de Cruzeiros do Porto de Leixões, Av. General Norton de Matos s/n, 4450-208 Matosinhos, Portugal; 3Department of Pharmacy, National and Kapodistrian University of Athens, 15771 Athens, Greece; 4Instituto de Biologia Molecular e Celular (i3S-IBMC), Universidade do Porto, Rua de Jorge Viterbo Ferreira, 228, 4050-313 Porto, Portugal; 5Department of Chemistry, University of Leicester, University Road, Leicester LE1 7RH, UK; 6HEJ Research Institute of Chemistry, International Center for Chemical and Biological Sciences, University of Karachi, Karachi 75270, Pakistan; 7Departamento de Química & QOPNA, Universidade de Aveiro, 3810-193 Aveiro, Portugal; 8Department of Plant Pathology, Faculty of Agriculture, Kasetsart University, Bangkok 10240, Thailand; 9Laboratório de Química Orgânica e Farmacêutica, Departamento de Ciências Químicas, Faculdade de Farmácia, Universidade do Porto and CIIMAR, Rua de Jorge Viterbo Ferreira 228, 4050-313 Porto, Portugal

**Keywords:** *Talaromyces pinophilus*, Trichocomaceae, marine sponge-associated fungus, hybrid oxyphenalenone, highly conjugated C_28_ steroid, azaphilone, antibacterial activity, antibiofilm activity

## Abstract

An undescribed hybrid phenalenone dimer, talaropinophilone (**3**), an unreported azaphilone, 7-*epi*-pinazaphilone B (**4**), an unreported phthalide dimer, talaropinophilide (**6**), and an undescribed 9*R*,15*S*-dihydroxy-ergosta-4,6,8 (14)-tetraen-3-one (**7**) were isolated together with the previously reported bacillisporins A (**1**) and B (**2**), an azaphilone derivative, Sch 1385568 (**5**), 1-deoxyrubralactone (**8**), acetylquestinol (**9**), piniterpenoid D (**10**) and 3,5-dihydroxy-4-methylphthalaldehydic acid (**11**) from the ethyl acetate extract of the culture of a marine sponge-derived fungus, *Talaromyces pinophilus* KUFA 1767. The structures of the undescribed compounds were elucidated by 1D and 2D NMR as well as high-resolution mass spectral analyses. The absolute configuration of C-9′ of **1** and **2** was revised to be 9′*S* using the coupling constant value between C-8′ and C-9′ and was confirmed by ROESY correlations in the case of **2**. The absolute configurations of the stereogenic carbons in **7** and **8** were established by X-ray crystallographic analysis. Compounds **1**,**2**, **4**–**8**, **10** and **11** were tested for antibacterial activity against four reference strains, viz. two Gram-positive (*Staphylococcus aureus* ATCC 29213, *Enterococcus faecalis* ATCC 29212) and two Gram-negative (*Escherichia coli* ATCC 25922, *Pseudomonas aeruginosa* ATCC 27853), as well as three multidrug-resistant strains, viz. an extended-spectrum β-lactamase (ESBL)-producing *E. coli*, a methicillin-resistant *S. aureus* (MRSA) and a vancomycin-resistant *E. faecalis* (VRE). However, only **1** and **2** exhibited significant antibacterial activity against both *S. aureus* ATCC 29213 and MRSA. Moreover, **1** and **2** also significantly inhibited biofilm formation in *S. aureus* ATCC 29213 at both MIC and 2xMIC concentrations.

## 1. Introduction

Fungi of the genus *Talaromyces* (Trichocomaceae) are important due not only to their ubiquitous nature, as they can be isolated from soil, plants, marine organisms and foods, but also to their great potential in agricultural applications as biological control agents [1] and to degrade agricultural waste [2]. Moreover, *Talaromyces* species also produce a variety of specialized metabolites such as alkaloids, peptides, polyketides, quinones, steroids and terpenoids, some of which exhibited relevant biological and pharmacological activities [3,4].

Concerning *T. pinophilus*, several strains of this fungus, isolated from different terrestrial sources, including plant endophytic and soil-derived strains, have been investigated in the past years. These strains produced a variety of different chemical classes of specialized metabolites with unique structural features and relevant biological activities. The culture extract of *T. pinophilus* FKI-3864, isolated from a soil sample, produced several dihydronaphthopyranone-containing compounds, also known as dinapinones, one of which displayed potent inhibition of triacylglycerol synthesis in intact mammalian cells [5]. The endophytic *T. pinophilus* F36CF, which was recovered from a secondary branch of a strawberry tree, furnished the antibiotic 3-*O*-methylfunicone, a benzoyl 4-pyrone derivative, together with herquiline B and siderophore ferrirubin [6] while the culture of the strain LT6, recovered from the rhizosphere of a tobacco plant, produced a macrocyclic lactone, named talarodiolide, and other previously reported compounds [7]. Zhao et al. reported the isolation of a tricyclic fusicoccane diterpene, pinophilin, and a polyene, pinophol, from the endophytic *T. pinophilus* XL-1193, isolated from the aerial part of *Salvia miltiorhiza* [8].

In our ongoing search for antibiotic and antibiofilm compounds from marine-derived fungi from tropical seas, we selected a marine-derived *T. pinophilus*, isolated from a marine sponge, *Mycale* sp., which was collected from the coral reef at Samaesan Island in Chonburi province, Thailand.

Fractionation of the ethyl acetate (EtOAc) extract of the culture of *T. pinophilus* KUFA 1767 by silica gel column chromatography, followed by purification by preparative TLC, Sephadex LH-20 column and crystallization, led to the isolation of an undescribed hybrid phenalenone dimer, talaropinophilone (**3**), 7-*epi*-pinazaphilone B (**4**), talaropinophilide (**6**) and 9*R*,15*S*-dihydroxy-ergosta-4,6,8 (14)-tetraen-3-one (**7**), together with the previously reported bacillisporins A (**1**) and B (**2**) [9], Sch 1385568 (**5**) [10,11], 1-deoxyrubralactone (**8**) [12], acetylquestinol (**9**) [13], piniterpenoid D (**10**) [14,15] and 3,5-dihydroxy-4-methyl phthalaldehydic acid (**11**) [16] (Figure 1).

In the case of 1-deoxyrubralactone (**8**) (Appendix A), we determined the absolute configuration of C-11 as 11*S* by single-crystal X-ray analysis. Moreover, the absolute configuration of C-9′ in bacillisporins A (**1**) and B (**2**) was corrected to be 9′*S* based on the lack (or very small quantity) of a coupling constant between H-8′ and H-9′ in both compounds. In the case of bacillisporin B (**2**), the configurations of its C-8′ and C-9′ were also supported by ROESY correlations.

## 2. Results and Discussion

The structures of bacillisporins A (**1**) and B (**2**) [9], Sch 1385568 (**5**) [10,11], 1-deoxyrubralactone (**8**) [12], acetylquestinol (**9**) [13], piniterpenoid D (**10**) [14,15] and 3,5-dihydroxy-4-methyl-phthalaldehydic acid (**11**) [16] were elucidated by the analysis of their 1D and 2D NMR spectral data (Appendix A) and by the comparison of their NMR spectral data and, in some cases, the optical rotation, with those reported in the literature.

The ^1^H and ^13^C NMR spectra of **1** (Appendix A) and **2** (Table 1, Appendix A), in DMSO-*d6*, were identical to those of bacillisporins A and B we had previously isolated from the culture of a soil fungus, *T. bacillisporus* R. Benjamin. Moreover, their optical rotations ([α]_D_^20^ + 24.2 (MeOH, *c* 0.050) for **1** and [α]_D_^20^ + 45 (MeOH, *c* 0.044) for **2**) are also very similar to those of the previously reported bacillisporins A and B ([α]**_D_^20^**+ 27.1 (MeOH, *c* 0.023) for bacillisporin A and [α]_D_^20^ + 31.8 (MeOH, *c* 0.022) for bacillisporins B) [9]. However, the absolute configuration at C-9′ of bacillisporins A and B in this paper was presented as 9′*R*, by the comparison of their ^1^H and ^13^C NMR chemical shift values and the signs of its optical rotation (dextrorotatory) with those of bacillisporins A and B previously reported by Yamazaki et al. [17].

It is important to mention that, during the process of structure elucidation of duclauxamide A1, an analog of duclauxin, from the culture extract of *Penicillium manginii* YIM PH30375, Cao et al. observed that H-8′ and H-9′ appeared as broad singlets in all structurally related duclauxin analogues, implying that the dihedral angle between C-8′ and C-9′ is ca. 90°, which corresponds to the 9′*S* configuration and not the 9′*R* configuration as previously proposed. This configuration was corroborated by the X-ray analysis of duclauxamide A1 using a CuKα X-ray diffractometer [18]. Since H-8′ and H-9′ of **1** appeared as a broad singlet at δ_H_ 4.99 and a doublet at δ_H_ 5.80 (*J* = 0.8 Hz), respectively, and those of **2** appeared as a singlet at δ_H_ 4.82 and a broad singlet at δ_H_ 4.76, respectively, we concluded that the absolute configuration at C-9′ in **1** and **2** is the same as that of C-9′ of duclauxamide, i.e., 9′*S*. In order to confirm this assumption, the ROESY spectrum of **2** was obtained. The ROESY spectrum (Table 1, Appendix A) showed correlations between H_3_-10 (δ_H_ 2.97, s) and H-5 (δ_H_ 6.95, s) and between H_3_-10′ (δ_H_ 2.48, s) and H-5′ (δ_H_ 6.82, d, *J* = 0.6 Hz). Moreover, H-8′ also showed correlation to H_3_-10, while H-9′ showed correlation to a doublet at δ_H_ 5.13, d, *J* = 12.4 Hz. Therefore, the doublet at δ_H_ 5.13 was assigned to H-1′α. Meanwhile, no ROESY correlation was observed from H-8′ to H-9′, implying that their dihedral angle is ca. 90° and not ca. 30°, and the configuration of C-9′ in **2** is the same as that of duclauxin, i.e., 9′*S*.

Compound **3** was isolated as a yellow amorphous solid, and its molecular formula C_27_H_18_O_8_ was established based on (+)-HRESIMS *m*/*z* 471.1075 [M + H]^+^ (calculated for C_27_H_19_O_8_, 471.1080) and 493.0895 [M + Na]^+^ (calculated for C_27_H_18_O_8_Na, 493.0899) (Appendix A), requiring nineteen degrees of unsaturation. The ^13^C NMR spectrum (Table 2, Appendix A) exhibited 27 carbon signals which, in combination with DEPT and HSQC spectra, can be categorized as one ketone carbonyl (δ_C_ 192.4), one conjugated ester carbonyl (δ_C_ 168.1), one highly conjugated ketone carbonyl (δ_C_ 159.3), thirteen non-protonated sp^2^ (δ_C_ 161.0, 157.3, 152.0, 151.0,148.1, 142.8, 137.9, 133.7, 130.3, 122.4, 117.0, 112.6, 104.5), five protonated sp^2^ (δ_C_ 141.2, 122.0, 121.1, 114.8, 103,9), one oxymethine sp^3^ (δ_C_ 86.0), one methine sp^3^ (δ_C_ 65.7), one oxymethylene sp^3^ (δ_C_ 70.1), one quaternary sp^3^ (δ_C_ 50.0) and two methyl singlets (δ_C_ 29.4 and 23.7), respectively. The ^1^H and ^13^C NMR (Table 2, Appendix A) spectral features of **3** resemble those of **2** (Table 1, Appendix A) except for the presence of two doublets of the olefinic protons at δ_H_ 8.60 (*J* = 10 Hz) and 6.46 (*J* = 10 Hz) and a conjugated ketone carbonyl (δ_C_ 159.3) in the former instead of two oxymethylene doublets (δ_H_ 5.72, d, *J* = 15.1 Hz and 5.64, d, *J* = 15.1 Hz, H_2_-1) and a conjugated ester carbonyl (δ_C_ 168.1) in the latter. That **3** contains the same 4-hydroxy-3,3a,4,5-tetrahydro-1*H*, 6*H*-naphtho [1,8-*cd*]pyran-1,6-dione (the dihydroisocoumarin moiety) as in **2** was evidenced by the presence of H-8′ at δ_H_ 4.97 brs (δ_C_ 65.7), H-9′ at δ_H_ 4.99 s (δ_C_ 86.0), H_2_-1′ at δ_H_ 5.04, d, *J* = 12.6 Hz and 5.17, d, *J* = 12.6 Hz (δ_C_ 70.1), H-5′ at δ_H_ 6.77s (δ_C_ 121.0), Me-10′ at δ_H_ 2.48, s (δ_C_ 23.7), C-3′ (δ_C_ 168.1) and C-7′ (δ_C_192.4). This structure was supported by the COSY, HSQC and HMBC correlations shown in Table 2 and Appendix A.

That another portion of **3** consists of 3-methylnaphthalen-1-ol was supported by the presence of two *meta*-coupled aromatic protons at δ_H_ 7.54, d, *J =* 2.0 Hz (H-4, δ_C_ 103.9) and δ_H_ 7.04, d, *J =* 2.0 Hz (H-6, δ_C_ 122.0), a methyl singlet at δ_H_ 3.04 (Me-10, δ_C_ 24.9), which exhibited COSY correlations to both H-4 and H-6 (Table 2, Appendix A). This partial structure was also supported by HMBC correlations between H-4 and the carbons at δ_C_ 157.3 (C-5), C-6 and the carbon at δ_C_ 122.4 (C-7a), H-6 to C-4, Me-10 and C-7a (Table 2, Appendix A). The existence of a pyran-4-one portion, which is fused with the benzene ring of the 3-methylnaphthalen-1-ol, was substantiated by, in addition to the presence of the two doublets at δ_H_ 8.60, d, *J* = 10 Hz (H-3) and 6.46, d, *J* = 10 Hz (H-2) of a *cis* double bond (Table 2, Appendix A), the HMBC correlations between H-2 and the carbons at δ_C_ 159.4 (C-1) and 112.6 (C-9a), while H-3 showed HMBC correlations to C-1 and the carbon at δ_C_ 151.0 (C-3a) (Table 2, Appendix A). Since H-8′ and H-9′ showed HMBC correlations to the carbons at δ_C_ 130.3 (C-9) and 142.8 (C-8) while H_2_-1′ showed only HMBC correlation to C-9, the 4-hydroxy-3,3a,4,5-tetrahydro-1*H*, 6*H*-naphtho [1,8-cd]pyran-1, 6-dione is linked to the 4*H*-naphtho [1,2-*b*]pyran-4-one moiety through C-8/C-8′ and C-9/C-9′a.

The ROESY spectrum (Table 2, Appendix A) showed correlations between H-3 and H-2 and H-4, confirming the position of the 4-pyrone ring. Moreover, the correlation between H-8′ and H_3_-10, together with the lack of correlation between H-8′ and H-9′, proves that the absolute configuration at C-9′ of **3** is the same as that of C-9′ of **2**, i.e., 9′*S.* Compound **3** is a heterodimer consisting of an oxyphenalenone and a naphthopyrone monomer. Since **3** has never been previously reported, it was named talaropinophilone.

Compound **3** is a polyketide whose biosynthesis can be assumed to originate from a dimerization of two different monomers: 6,9-dihydroxy-7-methyl-4*H*-naphtho [1,2-*b*]pyran-4-one (**IX**) and SF22, both of which are derived from heptaketides, in the same manner as that proposed for the formation of duclauxins by Cao et al. (Figure 2) [18]. Compound **IX** is derived from a heptaketide **I** by cyclization, through a nucleophilic addition of the ketone carbonyl, to form **II**, which undergoes dehydration and enolization to form a benzene ring in **III**. Enolization and reduction of a ketone carbonyl in **III** generate a secondary alcohol in **IV**. Dehydration of the secondary alcohol in **IV** results in the formation of **V**, which is followed by a Claisen condensation to form **VI**. Aromatic hydroxylation of **VI** gives rise to **VII**, which undergoes a hemiacetal formation in **VIII**. Dehydration of **VIII** gives rise to a monomer **IX**. On the other hand, the monomer SF 226 is also proposed to be derived from a heptaketide through lamellicolic anhydride [18].

Compound **4** was isolated as a yellow viscous mass, and its molecular formula C_27_H_18_O_8_ was established based on (+)-HRESIMS *m*/*z* 417.1186 [M + H]^+^ (calculated for C_21_H_21_O_9_, 417.1186) (Appendix A), requiring twelve degrees of unsaturation. Analysis of the 1D and 2D NMR spectra (Table 3, Appendix A) of **4** revealed that its planar structure is the same as that of pinazaphilone B, a benzoyl-substituted azaphilone, isolated from a solid culture extract of an endophytic fungus, *Penicillium* sp. HN29-3B1, which was obtained from a fresh branch of a mangrove tree, *Cerbera manghas*, collected from Dongzhaigang Mangrove National Nature Reserve in Hainan Island [11]. In fact, the ^1^H and ^13^C NMR data of **4**, recorded in DMSO-*d6* (Table 3, Appendix A), were similar to those of pinazaphilone B (recorded in acetone-*d6*) with some slightly different ^1^H and ^13^C chemical shift values of the benzoyl moiety as well as of Me-9. Intriguingly, the optical rotation of **4** was dextrorotatory ([α]^25^_D_ + 40 (*c* 0.05, MeOH)), whereas that of pinazaphilone B was levorotatory ([α]^25^_D_ − 50 (*c* 0.001, MeOH)). This prompted us to examine the stereochemistry of the stereogenic carbons (C-7, C-8 and C-8a) of **4**. Since the value of the coupling constant of *J*_8−8a_ (9.8 Hz) in **4** is the same as that in pinazaphilone B, indicating a *trans*-diaxial relationship between H-8 and H-8a, the absolute configurations of C-8 and C-8a must be the same in both compounds. The same stereochemistry at C-8a for both compounds was corroborated by the same values of the coupling constants of H-1α (δ_H_ 3.85, dd, *J* = 13.5, 11.0 Hz) and H-1β (δ_H_ 4.41, dd, *J* = 11.0, 5.2 Hz) of **4** and of H-1α (δ_H_ 3.94, dd, *J* = 13.5, 10.9 Hz) and H-1β (δ_H_ 4.61, dd, *J* = 10.9, 5.2 Hz) of pinazaphilone B.

Interestingly, examination of correlations in the ROESY spectrum of **4** (Table 3, Appendix A) revealed strong cross peaks from H-8 (δ_H_ 5.12, d, *J* = 9.8 Hz) to Me-9 (δ_H_ 1.22, s) and H-1α. Thus, the configuration of C-7 in **4** is opposite to that of C-7 in pinazaphilone B, i.e., 7*S*. Therefore, **4** is an epimer of pinazaphilone B at C-7, and it was named 7-*epi*-pinazaphilone B. It is interesting to note that the absolute configuration at C-7 in **4** is the same as that of C-7 of Sch 1385568 (**5**), which was co-isolated with **4** in this work, and whose configuration at C-7 was confirmed by a ROESY correlation between Me-9 and H-8 (Appendix A). Compound **5** was previously isolated from a culture of *Aspergillus* sp. [19], as well as together with pinazaphilone B from the culture of *Penicillium* sp. HN29-3B1 [11]. To the best of our knowledge, this is the first report of 7-*epi*-pinazaphilone B (**4**).

Compound **6** was isolated as a white solid (mp. 230–231 °C). Its molecular formula C_19_H_16_O_9_ was established based on (+)-HRESIMS *m*/*z* 389.0807[M + H]^+^, calculated for C_19_H_17_O_9_, 389.0873 (Appendix A), requiring twelve degrees of unsaturation. The ^13^CNMR spectrum (Table 4, Appendix A) displayed 19 carbon signals which, in combination with DEPT and HSQC spectra (Appendix A), can be categorized as two conjugated ester carbonyls (δ_C_ 171.7 and 169.0), four oxygen-bearing sp^2^ (δ_C_ 158.8, 155.2, 152.3 and 148.2), seven non-protonated sp^2^ (δ_C_ 126.0, 125.2, 122.6, 121.6, 121.4, 118.9, 117.0), two protonated sp^2^ (δ_C_ 102.8 and 101.4), one oxymethylene sp^3^ (δ_C_ 67.0), one methoxyl (δ_C_ 56.6), one methylene sp^3^ (δ_C_ 18.3) and one methyl (δ_C_ 10.2) carbons. Analysis of the HMBC spectrum (Table 4, Appendix A) revealed that **6** contains a 5-substituted 4,6-dihydroxy-3-methoxy-2-benzofuran-1-(3*H*)-one moiety. This was supported by HMBC correlations between H-6 (δ_H_ 6.69, s) and the conjugated ester carbonyl at δ_C_ 169.0 (C-7), the carbon signals at δ_C_ 158.6 (C-5), 122.6 (C-1) and 121.4 (C-4), between a singlet of an acetal proton at δ_H_ 6.29 (H-8) and C-7 and the methoxyl carbon at δ_C_ 56.6, as well as between the methoxyl proton (δ_H_ 3.44, s) and the acetal carbon at δ_C_ 102.8 (C-8) (Table 4, Appendix A). That another portion of the molecule is 4,6-dihydroxy-5-methyl-2-benzofuran-1(3*H*)-one was evidenced by HMBC correlations between an oxymethylene singlet at δ_H_ 5.14 (H-18) and the conjugated ester carbonyl at δ_C_ 171.7 (C-17), a substituted aromatic carbon at δ_C_ 125.2 (C-15) and a hydroxyl-bearing aromatic carbon at δ_C_ 148.2, as well as between a methyl singlet at δ_H_ 2.04 (H-19; δ_C_ 10.2) and C-14, a hydroxyl-bearing aromatic carbon at δ_C_ 155.2 (C-12) and a substituted aromatic carbon at δ_C_ 118.9 (C-13) (Table 4, Appendix A). That the 5-substituted 4,6-dihydroxy-3-methoxy-2-benzofuran-1-(3*H*)-one moiety is connected to the 4,6-dihydroxy-5-methyl-2-benzofuran-1(3*H*)-one moiety by a methylene bridge was evidenced by the COSY correlations between H-9a (δ_H_ 4.35, d, *J* = 14.4 Hz) and H-9b (δ_H_ 4.47, d, *J* = 14.4 Hz) and H-6 and H-18 (Table 4, Appendix A) as well as HMBC correlations between H-9a/H-9b and C-4, C-5, C-11 (δ_C_ 121.6), C-12 and C-3 (δ_C_ 152.3) (Table 4, Appendix A). Therefore, the planar structure of **6** was established. Compound **6** contains a stereogenic carbon at C-8 and displayed an optical rotation ([α]_D_^20^ − 16 (*c* 0.05, MeOH)). It is important to mention that a few fungal specialized metabolites containing 5-substituted-4,6-dihydroxy-3-methoxy-2-benzofuran-1-(3*H*)-one have been previously reported. Kimura et al. [16] reported the isolation of rubralides A and C from a culture extract of a soil fungus, *Penicillium rubrum*, while Zhai et al. [20] described the isolation of talaromycolides A-C as well as rubralide C. Although all of these compounds have the same absolute configuration of the stereogenic carbon (C-8) and the same 4,6-dihydroxy-3-methoxy-2-benzofuran-1-(3*H*)-one scaffold but differ in the substituents on C-4, they showed different signs of optical rotations. For example, rubralides A and C and talaromycolide A are dextrorotatory, whereas talaromycolides B and C are levorotatory. Interestingly, talaromycolides A, B and C have benzyl substituents on C-5 but showed opposite signs of rotations. Therefore, the optical rotation cannot be used to determine the absolute configuration of the stereogenic carbon of this series of the 3-methoxy-2-benzofuran-1-(3*H*)-one scaffold. However, Zhai et al. [20] determined the absolute configuration of the stereogenic carbons in talaromycolides A, B and C as *R* by comparing their CD curves (in ethanol) and the positive values of Δε (at 266 and 288 for talaromycolides A, 263 and 287 for talaromycolides B, 255 and 296 for talaromycolides C) with those of rubralide C (269 and 288).

Intriguingly, **6** in MeOH solution (as it is not completely soluble in ethanol) did not display any CD curve even at a concentration of as high as 5 mg/mL. Consequently, it was not possible to determine the absolute configuration at C-8 of **6**. However, this phenomenon reflects the possibility that **6** could exist as a non-racemic mixture of both enantiomers or could be due to a high similarity of the two phthalide monomers. Since **6** has never been previously reported, it was named talaropinophilide.

Compound **7** was isolated as white crystals, and its molecular formula C_28_H_40_O_3_ was established based on (+)-HRESIMS *m*/*z* 425.3053 [M + H]^+^ (calculated for C_28_H_41_O_3_, 425.3056) (Appendix A), requiring nine degrees of unsaturation. The ^13^C NMR spectrum (Table 5, Appendix A) displayed 28 carbon signals which, in combination with DEPT and HSQC spectra (Appendix A), can be classified as one ketone carbonyl (δ_C_ 198.6), three non-protonated sp^2^ (δ_C_ 163.4, 158.2, 131.3), five protonated sp^2^ (δ_C_ 135.5, 132.6, 132.3, 124.9 and 124.8), one oxyquaternary sp^3^ (δ_C_ 71.8), two quaternary sp^3^ (δ_C_ 44.5 and 42.0), one oxymethine sp^3^ (δ_C_ 68.4), four methine sp^3^ (δ_C_ 52.5, 42.6, 38.5, 33.0), five methylene sp^3^ (δ_C_ 40.3, 34.0, 31.7, 27.5, 26.4), two tertiary (δ_C_ 19.3 and 21.0) methyl and four secondary methyl (δ_C_ 21.7, 20.3, 19.9, and 17.9) carbons. The numbers and types of the carbon atoms are characteristic of an ergostane-type skeleton. The ^1^H NMR spectrum (Table 5, Appendix A) exhibited signals of five olefinic protons, consisting of a singlet at δ_H_ 5.70 (δ_C_124.9), two doublets at δ_H_ 6.08 (*J =* 9.6 Hz, δ_C_124.8) and 7.04 (*J* = 9.6 Hz, δ_C_132.6) and two multiplets, centered at δ_H_ 5.24 (δ_C_ 135.5) and 5.25 (δ_C_ 132.5). The ^1^H and ^13^C NMR spectral features of **7** resemble those of ergosta-4,6,8 (14),22-tetraen-3-one, previously isolated by us from the culture extract of a marine sponge-associated fungus, *Neosartorya glabra* KUFA 0702 [21], except for the presence of one oxymethine and one oxyquaternary sp^3^ carbons **7** instead of one methine and one methylene sp^3^ carbons in ergosta-4,6,8 (14),22-tetraen-3-one. The molecular formula of **7** indicated that the hydroxyl groups are on the oxyquaternary sp^3^ (δ_C_ 71.8) and oxymethine sp^3^ (δ_C_ 68.4) carbons. That the hydroxyl group is on C-9 (δ_C_ 71.8) was confirmed by HMBC correlations between H-7 (δ_H_ 7.04, d, *J* = 9.6 Hz) and Me-18 (δ_H_ 1.04, s) and C-9 (Table 5, Appendix A). Another hydroxyl group is on C-15 due to a COSY correlation between H-15 (δ_H_ 4.64, m) and H_2_-16 (δ_H_ 1.69, m) (Table 5, Appendix A). Since both of the olefinic protons of the side chain appeared as multiplets, it was not possible to determine the configuration of the double bond of the side chain. Therefore, the planar structure of **7** was established as 9,15-dihydroxy-ergosta-4,6,8 (14)-tetraen-3-one, since **7** was obtained as a suitable crystal for X-ray analysis. The Ortep view of **7**, obtained from an X-ray diffractometer equipped with CuKα radiation (Figure 3), showed clearly that the absolute configurations at C-7 and C-15 are 9*R* and 15*S* and the double bond between C-22 and C-23 is *trans*. Therefore, the structure of **7** was unambiguously established as 9*R*,15*S*-dihydroxy-ergosta-4,6,8 (14)-tetraen-3-one. To the best of our knowledge, this is the first report of **7**.

Compound **8** was isolated as white crystals (mp = 199–201 °C). Analysis of the ^1^H and ^13^C NMR spectra, as well as COSY and HMBC correlations (Appendix A), revealed that the planar structure of **8** is the same as that of 1-deoxyrubralactone, isolated from cultures of the fungal strain HJ33moB, which was isolated from sea algae, collected in Hatijou Island, Japan [12]. However, Naganuma et al. did not determine the absolute configuration of C-11 (C-1 in the structure of 1-deoxyrubralactone in reference [12]). Since **8** was obtained as a suitable crystal for X-ray analysis using an X-ray diffractometer equipped with CuKα radiation, the configuration of C-11 was determined. The Ortep view of **8** is shown in Figure 4, revealing that the absolute configuration of C-11 is 11*S*. Interestingly, 1-deoxyrubralactone isolated by Naganuma et al. [12] was levorotatory, with [α]_D_^22^ − 8.33 (*c* 0.06, CHCl_3_), while **8** was dextrorotatory, with [α]_D_^20^ + 58.0 (*c* 0.003, MeOH).

The antimicrobial activity of **1**, **2**, **4**–**8**, **10** and **11** was evaluated against four reference bacterial species and three multidrug-resistant strains. Since **3** was obtained in a very small quantity and **9** has been previously tested for antimicrobial activity against the same strains, they were not included in these assays.

Compounds **1**, **2** and **5** exhibited inhibitory activity against the tested strains (Table 6). Compound **1** showed very potent activity against both reference and multidrug-resistant *Staphylococcus* strains (*S. aureus* ATCC 29213 and MRSA *S. aureus* 74/24), with MIC values of 4 µg/mL. Compound **2** had an MIC value of 8 and 16 µg/mL for *S. aureus* ATCC 29213 and MRSA *S. aureus* 74/24, respectively, and 32 µg/mL for *E. faecalis* ATCC 29212. Compound **5** was also able to inhibit the growth of *S. aureus* ATCC 29213, with an MIC value of 64 µg/mL. For the remaining compounds that were tested, no activity was detected at concentrations up to 64 µg/mL. 

The minimum bactericidal concentration (MBC) was not possible to determine in the range of concentrations tested (higher than 64 µg/mL), except for **1**, **2** and **5**, where the MBC was more than two-fold higher than the MIC, suggesting that they have a bacteriostatic effect.

The potential synergy between the compounds and clinically relevant antibiotics on multidrug-resistant strains was evaluated. No interactions were found with cefotaxime (ESBL *E. coli*), methicillin (MRSA *S. aureus*) and vancomycin (VRE *E. faecalis*) with both disk diffusion and MIC methods. Thus, the checkerboard assay was not further performed for any compound. The MIC values of clinically relevant antibiotics for multidrug-resistant strains were 256 µg/mL of cefotaxime for ESBL *E. coli* SA/2, 64 µg/mL of oxacillin for MRSA *S. aureus* 74/24 and 1024 µg/mL of vancomycin for VRE *E. faecalis* B3/101.

The ability of **1**, **2**, **4**–**8**, **9** and **10** to prevent biofilm production was also evaluated, in all reference strains, by measuring the total biomass. Compounds that showed antibiofilm activity after 24 h incubation are shown in Table 7. Compounds **1** and **2** showed an extensive ability to significantly inhibit biofilm formation in *S. aureus* ATCC 29213 at both MIC and 2xMIC concentrations. Compound **2** was also capable of impairing the biofilm-forming ability of *E. faecalis* ATCC 29212 at 2xMIC concentration.

As for the overall effect of the compounds, **1** and **2** showed promising results in terms of antibacterial and antibiofilm activities toward Gram-positive bacterial strains, requiring a more in-depth study of their mechanism by which they inhibit bacterial growth and biofilm production.

## 3. Experimental Sections

### 3.1. General Experimental Procedures

The melting points were determined on a Stuart Melting Point Apparatus SMP3 (Bibby Sterilin, Stone, Staffordshire, UK) and are not corrected. Optical rotations were measured on an ADP410 Polarimeter (Bellingham + Stanley Ltd., Tunbridge Wells, Kent, UK). ^1^H and ^13^C NMR spectra were recorded at ambient temperature on a Bruker AMC instrument (Bruker Biosciences Corporation, Billerica, MA, USA) operating at 300 or 500 and 75 or 125 MHz, respectively. High-resolution mass spectra were measured with a Waters Xevo QToF mass spectrometer (Waters Corporations, Milford, MA, USA) coupled to a Waters Aquity UPLC system. A Merck (Darmstadt, Germany) silica gel GF_254_ was used for preparative TLC, and a Merck Si gel 60 (0.2–0.5 mm) was used for column chromatography. LiChroprep silica gel and Sephadex LH 20 were used for column chromatography.

### 3.2. Fungal Material

The fungus was isolated from a marine sponge, *Mycale* sp., which was collected by scuba diving at a depth of 5–10 m, from the coral reef at Samaesan Island (12°34′36.64″ N 100°56′59.69″ E), in the Gulf of Thailand, Chonburi Province, in May 2018. The sponge was washed with 0.01% sodium hypochlorite solution for 1 min, followed by sterilized seawater three times, and then dried on a sterile filter paper under sterile aseptic condition. The sponge was cut into small pieces (*ca*. 5 mm × 5 mm) and placed on Petri dish plates containing 15 mL potato dextrose agar (PDA) medium mixed with 300 mg/L of streptomycin sulfate and incubated at 28 °C for 7 days. The hyphal tips emerging from sponge pieces were individually transferred onto PDA slant and maintained as pure cultures at Kasetsart University Fungal Collection, Department of Plant Pathology, Faculty of Agriculture, Kasetsart University, Bangkok, Thailand. The fungal strain KUFA 1767 was identified as *Talaromyces pinophilus*, based on morphological characteristics such as colony growth rate and growth pattern on standard media, namely Czapek′s agar, Czapek yeast autolysate agar and malt extract agar. Microscopic characteristics including size, shape and ornamentation of conidiophores and spores were examined under light microscope. This identification was confirmed by molecular techniques using internal transcribed spacer (ITS) primers. DNA was extracted from young mycelia following a modified Murray and Thompson method [22]. Primer pairs ITS1 and ITS4 were used for ITS gene amplification [23]. PCR reactions were conducted on Thermal Cycler, and the amplification process consisted of initial denaturation at 95 °C for 5 min, 34 cycles at 95 °C for 1 min (denaturation), at 55 °C for 1 min (annealing) and at 72 °C for 1.5 min (extension), followed by final extension at 72 °C for 10 min. PCR products were examined by agarose gel electrophoresis (1% agarose with 1 × Tris-Borate-EDTA (TBE) buffer) and visualized under UV light after staining with ethidium bromide. DNA sequencing analyses were performed using the dideoxyribonucleotide chain termination method [24] by Macrogen Inc. (Seoul, Republic of South Korea). The DNA sequences were edited using FinchTV software, submitted to the BLAST program for alignment and compared with that of fungal species in the NCBI database (http://www.ncbi.nlm.nih.gov/, accessed on 20 August 2019). Its gene sequences were deposited in GenBank with the accession number MZ331806.

### 3.3. Extraction and Isolation

The fungus was cultured for one week at 28 °C in five Petri dishes (i.d. 90 mm) containing 20 mL of potato dextrose agar per dish. The mycelial plugs (5 mm in diameter) were transferred to two 500 mL Erlenmeyer flasks containing 200 mL of potato dextrose broth and incubated on a rotary shaker at 120 rpm at 28 °C for one week. Fifty 1000 mL Erlenmeyer flasks, each containing 300 g of cooked rice, were autoclaved at 121 °C for 15 min. After cooling to room temperature, 20 mL of a mycelial suspension of the fungus was inoculated per flask and incubated at 28 °C for 30 days, after which 500 mL of EtOAc was added to each flask of the moldy rice, macerated for 7 days and then filtered with a Whatman No. 1 filter paper. The EtOAc solutions were combined and concentrated under reduced pressure to yield 95.1 g of a crude EtOAc extract which was dissolved in 500 mL of CHCl_3_, washed with H_2_O (3 × 500 mL), dried with anhydrous Na_2_SO_4_, filtered and evaporated under reduced pressure to give 63.3 g of a crude CHCl_3_ extract. The crude CHCl_3_ extract (60.0 g) was applied on a silica gel column (410 g) and eluted with mixtures of petrol-CHCl_3_ and CHCl_3_-Me_2_CO, wherein 250 mL fractions (frs) were collected as follows: frs 1–78 (petrol-CHCl_3_, 1:1), 79–146 (petrol-CHCl_3_, 3:7), 147–253 (petrol-CHCl_3_, 1:9), 254–414 (CHCl_3_–Me_2_CO, 9:1), 415–497 (CHCl_3_–Me_2_CO, 7:3), 498–527 (CHCl_3_–Me_2_CO, 1:1). Frs 93–102 were combined (1.72 g) and applied over a Sephadex LH-20 column (15 g) and eluted with MeOH, wherein 26 subfractions (sfrs) of 1 mL were collected. Ssfr 25–26 were combined (131.3 mg) and precipitated in MeOH to give 12.8 mg of **8**. Frs 156–164 were combined (756.7 mg) and applied over a Sephadex LH-20 column (15 g) and eluted with MeOH, wherein 11 sfrs of 3 mL were collected. Sfrs 9–11 were combined (66.9 mg) and applied over another Sephadex LH-20 column (5 g) and eluted with CHCl_3_, wherein 12 sub-subfractions (ssfrs) of 1 mL were collected. Ssfrs 5–8 were combined to give 16 mg of **1**. Frs 171–189 were combined (409 mg) and applied over a Sephadex LH-20 column (15 g) and eluted with MeOH, wherein 22 sfrs of 2 mL were collected. Sfrs 12–20 were combined (43.2 mg) and applied over another Sephadex LH-20 column (5 g) and eluted with CHCl_3_, wherein 23 ssfrs of 1 mL were collected. Ssfrs 10–18 were combined (4.6 mg) and precipitated in MeOH to give 2.5 mg of **9**. Frs 190–227 were combined (756 mg) and applied over a Sephadex LH-20 column (15 g) and eluted with MeOH, wherein 18 sfrs of 2 mL were collected. Sfrs 11–18 were combined (104.6 mg) and purified by preparative TLC (silica gel G_254_, CHCl_3_: petrol: Me_2_CO: HCO_2_H, 90:1:9:0.1) to give 10.0 mg of **5**. Frs 261–265 were combined (2.23 g) and applied over a Sephadex LH-20 column (15g) and eluted with MeOH, wherein 29 sfrs of 2 mL were collected. Sfrs 21–24 were combined (759.2 mg) and precipitated in MeOH to give 62.2 mg of **2**. Frs 280–286 were combined (256.1 mg) and applied over a Sephadex LH-20 column (15 g) and eluted with MeOH, wherein 17 sfrs of 2 mL were collected. Sfrs 2–11 were combined (127.5 mg) and purified by preparative TLC (silica gel G_254_, CHCl_3_: Me_2_CO: HCO_2_H, 90:9:0.1) to give 18.4 mg of **7**. Frs 294–336 were combined (1.1 g) and applied on a silica gel column (36 g) and eluted with mixtures of petrol-CHCL_3_ and CHCl_3_-Me_2_CO, wherein 100 mL sfrs were collected as follows: sfrs 1–11 (petrol-CHCl_3_, 3:7), 12–54 (petrol-CHCl_3_, 1:4), 55–86 (petrol-CHCl_3_, 1:9), 87–202 (CHCl_3_), 203–248 (CHCl_3_–Me_2_CO, 9:1), 249–267 (CHCl_3_–Me_2_CO, 7:3), 268–331 (CHCl_3_–Me_2_CO, 1:1), 332–344 (CHCl_3_–Me_2_CO, 3:7). Sfrs 215–248 were combined (167.6 mg) and applied over a Sephadex LH-20 column (15 g) and eluted with MeOH, wherein 30 ssfrs of 0.5 mL were collected. Sfrs 22–29 were combined (100.4 mg) and purified by preparative TLC (silica gel G_254_, CHCl_3_: Me_2_CO:HCO_2_H, 85:15:0.1) to give 2.5 mg of **3**. Frs 337–382 were combined (542.5 mg) and applied over a Sephadex LH-20 column (15 g) and eluted with MeOH, wherein 25 sfrs of 2 mL were collected. Sfr 14–16 were combined (84.1 mg) and precipitated in MeOH to give 6.7 mg of **10**. Frs 440–456 were combined (432.8 mg) and applied over a Sephadex LH-20 column (15 g) and eluted with MeOH, wherein 15 sfrs of 3 mL were collected. Sfrs 9–11 (63.6 mg) were combined and purified by preparative TLC (silica gel G_254_, CHCl_3_: Me_2_CO: HCO_2_H, 80:20:0.1) to give 22.9 mg of **4**. Frs 457–500 were combined (761.7 mg) and applied over a Sephadex LH-20 column (15 g) and eluted with MeOH, wherein 16 sfrs of 3 mL were collected. Sfrs 13–14 were combined (38.4 mg) and applied over another Sephadex LH-20 column (5 g) and eluted with CHCl_3_, wherein 17 ssfrs of 2 mL were collected. Ssfrs 10–12 were combined (17.8 mg) and precipitated in MeOH to give 15.5 mg of **6**. Frs 501–527 were combined (705.5 mg) and applied over a Sephadex LH-20 column (15 g) and eluted with MeOH, wherein 17 sfrs of 3 mL were collected. Sfr 9–12 were combined (87.8 mg) and precipitated in CHCl_3_ to give 15.8 mg of **11**. 

#### 3.3.1. Talaropinophilone (**3**)

Yellow amorphous solid. [α${]}_{D}^{20}$ +45 (*c* 0.044, MeOH); for ^1^H and ^13^C spectroscopic data (DMSO-*d6*, 300 and 75 MHz), see Table 2; (+)-HRESIMS *m*/*z* 471.1075 [M + H]^+^ (calculated for C_27_H_19_O_8_, 471.1080) and 493.0895 (M + Na)^+^ (calculated for C_27_H_28_O_8_Na, 493.0899).

#### 3.3.2. 7-Epi-pinazaphilone B (**4**)

Yellow viscous mass. [α${]}_{D}^{20}$ +40 (*c* 0.05, MeOH); for ^1^H and ^13^C spectroscopic data (DMSO-*d6*, 300 and 75 MHz), see Table 3; (+)-HRESIMS *m*/*z* 417.1186 [M + H]^+^ (calculated for C_21_H_21_O_9_, 417.1186).

#### 3.3.3. Talaropinophilide (**6**)

White solid. Mp. 230–231^o^C. [α${]}_{D}^{20}$ −16 (*c* 0.05, MeOH); for ^1^H and ^13^C spectroscopic data (DMSO-*d6*, 300 and 75 MHz), see Table 4; (+)-HRESIMS *m*/*z* 389.0807 [M + H]^+^ (calculated for C_19_H_17_O_9_, 389.0873).

#### 3.3.4. 9 R,15S-Dihydroxy-ergosta-4,6,8 (14)-tetraen-3-one (**7**)

White crystal. Mp. 134–136 °C. [α${]}_{D}^{20}$ +224 (*c* 0.041, MeOH); for ^1^H and ^13^C spectroscopic data (DMSO-*d6*, 300 and 75 MHz), see Table 5; (+)- HRESIMS *m*/*z* 425.3053 (M + H)^+^ (calculated for C_28_H_41_O_3_) and 425.3056; 447.2870 [M + Na]^+^ (calculated for C_28_H_40_O_3_Na, 447.2875).

### 3.4. X-ray Crystal Structures

Single crystals were mounted on a cryoloop using paratone. X-ray diffraction data were collected at 291 K with a Gemini PX Ultra equipped with CuK_α_ radiation (λ = 1.54184 Å). The structures were solved by direct methods using SHELXS-97 and refined with SHELXL-97 [25].

Full details of the data collection and refinement and tables of atomic coordinates, bond lengths and angles, and torsion angles have been deposited at the Cambridge Crystallographic Data Centre.

#### 3.4.1. X-ray Crystal Structure of **7**

The crystal was monoclinic, space group P2_1_, cell volume 2500.66(16) Å^3^ and unit cell dimensions *a* = 9.3410(3) Å, *b* = 19.7351(8) Å, *c* = 13.8316(5) Å and *β* = 101.265(3) (uncertainties in parentheses). Calculated crystal density was 1.122 g.cm^−3^. Non-hydrogen atoms were refined anisotropically. Hydrogen atoms were placed at their idealized positions using appropriate HFIX instructions in SHELXL and included in subsequent refinement cycles. The refinement converged to R (all data) = 11.55% and wR2 (all data) = 19.12%, and Flack parameter = 0.0(3). CCDC deposition number 2241767.

#### 3.4.2. X-ray Crystal Structure of **8**

The crystal was monoclinic, space group P2_1_, cell volume 595.19(3) Å^3^ and unit cell dimensions *a* = 9.6921(3) Å, *b* = 6.65370(10) Å, *c* = 10.4138(3) Å and *β* = 117.592(4) (uncertainties in parentheses). Calculated crystal density was 1.452 g.cm^−3^. Non-hydrogen atoms were refined anisotropically. Hydrogen was directly found from difference Fourier maps and refined freely with isotropic displacement parameters. The refinement converged to R (all data) = 3.80% and wR2 (all data) = 9.33%, and Flack parameter = 0.2(3). CCDC deposition number 2241769.

### 3.5. Electronic Circular Dichroism (ECD)

The attempt to obtain the ECD spectrum of **6** (with concentrations of 3 mg/mL and 5 mg/mL in methanol) was performed in a Jasco J-815 CD spectropolarimeter (Jasco Europe S.R.L., Cremella, Italy) with a 0.1 mm cuvette and 10 accumulations, but no CD curve was observed.

### 3.6. Antibacterial Activity Bioassays

#### 3.6.1. Bacterial Strains and Testing Conditions

Two Gram-positive (*Staphylococcus aureus* ATCC 29,213 and *Enterococcus faecalis* ATCC 29212) and two Gram-negative (*Escherichia coli* ATCC 25,922 and *Pseudomonas aeruginosa* ATCC 27853) reference strains, as well as three multidrug-resistant strains, were investigated: an extended-spectrum β-lactamase (ESBL)-producing *E. coli* (clinical isolate SA/2), a methicillin-resistant *Staphylococcus aureus* (MRSA, environmental isolate *S. aureus* 74/24 [26] and a vancomycin-resistant *Enterococcus* (VRE, environmental isolate *E. faecalis* B3/101 [27] were used in this study. All bacterial strains were cultured in MH agar (MH-BioKar Diagnostics, Allone, France) and incubated overnight at 37 °C before each susceptibility assay. Stock solutions of each compound were prepared in dimethyl sulfoxide (DMSO-Alfa Aesar, Kandel, Germany) at 10 mg/mL and kept at −20 °C. In all experiments, in-test concentrations of DMSO were kept below 1% (*v*/*v*), as recommended by the Clinical and Laboratory Standards Institute [28].

#### 3.6.2. Antimicrobial Susceptibility Testing

The antimicrobial activity of the compounds was assessed by the Kirby–Bauer method, according to CLSI recommendations [29]. Briefly, 6 mm diameter sterile blank paper discs (Oxoid/Thermo Fisher Scientific, Basingstoke, UK) were impregnated with 15 µg of each compound and placed on MH plates previously inoculated with a bacterial inoculum equal to 0.5 McFarland turbidity. After 18–20 h of incubation at 37 °C, the diameter of the inhibition zones was measured in mm. Blank paper discs impregnated with DMSO at 1% (*v*/*v*) concentration were used as a negative control. Minimum inhibitory concentrations (MICs) were determined by the broth microdilution method in a 96-well U-shaped untreated polystyrene, as recommended by the CLSI [30]. Two-fold serial dilutions of the compounds were prepared in cation-adjusted Mueller Hinton broth (CAMHB-Sigma-Aldrich, St. Louis, MO, USA), and the tested concentrations ranged from 1 to 64 µg/mL, in order to keep in-test concentrations of DMSO below 1%. Colony-forming unit (CFU) counts of the inoculum were conducted to ensure that the final inoculum size closely approximated 5 × 10^5^ CFU/mL. The 96-well plates were incubated for 16–20 h at 37 °C, and the MIC was determined by visual inspection as the lowest concentration of compound that prevented visible growth. During the essays, vancomycin (VAN-Oxoid/Thermo Fisher Scientific, Basingstoke, UK) and oxacillin sodium salt monosulfate (OXA-Sigma-Aldrich, St. Louis, MO, USA) were used as positive controls for *E. faecalis* ATCC 29212 and *S. aureus* ATCC 29213, respectively. The minimum bactericidal concentration (MBC) was determined by spreading 10 µL of the content of the wells with no visible growth on MH plates. The MBC was defined as the lowest concentration to effectively reduce 99.9% of the bacterial growth after overnight incubation at 37 °C [31]. At least three independent assays were conducted for reference and multidrug-resistant strains.

#### 3.6.3. Antibiotic Synergy Testing

The combining effect of each compound with clinically relevant antibacterial drugs was evaluated by Kirby–Bauer method, as previously described [32]. A set of antibiotic discs (Oxoid/Thermo Fisher Scientific, Basingstoke, UK) was selected based on the resistance of the multidrug-resistant strains toward these antibiotics: cefotaxime (CTX, 30 µg) for *E. coli* SA/2, vancomycin (VAN, 30 µg) for *E. faecalis* B3/101 and oxacillin (OXA, 1 µg) for *S. aureus* 74/24. Antibiotic discs impregnated with 15 µg of each compound were placed on seeded MH plates. Antibiotic discs alone, blank paper discs impregnated with 15 µg of each compound alone and blank discs impregnated with DMSO were used as controls in the experiment. Plates with CTX were incubated for 18–20 h, and plates with VAN and OXA were incubated for 24 h at 37 °C [28]. Potential synergy was considered when the inhibition halo of an antibiotic disc impregnated with compound was greater than the inhibition halo of the antibiotic or compound-impregnated blank disc alone. The MIC method was also performed in order to evaluate the combined effect of the compounds and clinically relevant antimicrobial drugs. Briefly, when it was not possible to determine the MIC value for the tested compound, the MIC of CTX (Duchefa Biochemie, Haarlem, The Netherlands), VAN (Oxoid, Basingstoke, UK) and OXA (Sigma-Aldrich, St. Louis, MO, USA) for the respective multidrug-resistant strain was determined in the presence of the highest concentration of each compound tested in previous assays (64 µg/mL). The tested antibiotic was serially diluted, whereas the concentration of each compound was kept fixed. MICs of antibiotics were determined as described above. Potential synergy was considered when the MIC of antibiotic was lower in the presence of the compound [33]. The checkerboard assay was only performed on compounds that had demonstrated synergistic potential in the previous methods, where fractional inhibitory concentrations (FICs) were calculated as follows: FIC of compound = MIC of compound combined with antibiotic/MIC compound alone, and FIC antibiotic = MIC of antibiotic combined with compound/MIC of antibiotic alone. The FIC index (FICI) was calculated as the sum of each FIC and interpreted as follows: FICI ≤ 0.5, ‘synergy’; 0.5 < FICI ≤ 4, ‘no interaction’; 4 < FICI, ‘antagonism’ [34].

#### 3.6.4. Biofilm Formation Inhibition Assay

The effect of compound on biofilm formation was evaluated by crystal violet method, which quantifies the total biomass produced, as previously described [32,35]. Briefly, bacterial suspensions of 1 × 10^6^ CFU/mL prepared in unsupplemented Tryptone Soy Broth (TSB-Biokar Diagnostics, Allone, Beauvais, France) or TSB supplemented with 1% (p/v) glucose (D-(+)-glucose anhydrous for molecular biology, PanReac AppliChem, Barcelona, Spain) were tested in a concentration range of 2 MIC and ¼ MIC, while keeping in-test concentrations of DMSO below 1% (*v*/*v*). When it was not possible to determine the MIC, the highest concentration tested was used (64 µg/mL). Controls with DMSO at 1% (*v*/*v*), as well as a negative control (TSB or TSB+1% glucose alone), were included. Sterile 96-well flat-bottomed untreated polystyrene microtiter plates were used. After a 24 h incubation period at 37 °C, the biofilms were heat-fixed for 1 h at 60 °C and stained with 0.5% (*v*/*v*) crystal violet (Química Clínica Aplicada, Amposta, Spain) for 5 min. The stain was resolubilized with 33% (*v*/*v*) acetic acid (Acetic acid 100%, AppliChem, Darmstadt, Germany) to completely solubilize the crystal violet. The absorbance of each sample was quantified at 570 nm in a microplate reader (Thermo Scientific Multiskan^®^ FC, Thermo Fisher Scientific, Waltham, MA, USA). The background absorbance (TSB or TSB + 1% glucose without inoculum) was subtracted from the absorbance of each sample, and the data are presented as percentage of control. At least three independent assays were performed for reference strains, with triplicates for each experimental condition.

## 4. Conclusions

A marine-derived fungus, *Talaromyces pinophilus* KUFA 1706, isolated from a marine sponge, *Mycale* sp., from the Gulf of Thailand, was shown to be a rich source of structurally diverse specialized metabolites. From the solid rice culture extract of this strain, one undescribed hybrid phenalenone dimer (**3**), one unreported azaphilone (**4**), one unreported phthalide dimer (**6**) and one unreported highly conjugated dihydroxy ergostane- type steroid (**7**) were isolated, together with the previously described oxyphenalenone dimers, bacillisporins A (**1**) and B (**2**) and the previously reported azaphilone (**5**), anthraquinone (**9**), aromatic cadinane sesquiterpenoid (**10**) and phthalic acid aldehyde (**11**). The biosynthetic pathways of the undescribed hybrid phenalenone dimer (**3**) were proposed to originate from heptaketide monomers, similar to those of the duclauxins. Among the compounds tested, only bacillisporins A (**1**) and B (**2**) exhibited significant antibacterial activity against both reference and multidrug-resistant *Staphylococcus* strains (*S. aureus* ATCC 29213 and MRSA *S. aureus* 74/24), as well as their ability to inhibit biofilm formation in *S. aureus* ATTC 29213. Interestingly, bacillisporin A (**1**), which possesses an acetoxy group at C-9′, showed stronger antibacterial activity and biofilm inhibitory activity than a hydroxyl-bearing C-9′ bacillisporin B (**2**). These results imply that the dimeric oxyphenalenone scaffold is essential for the antibacterial and antibiofilm activity while the nature of the substituents at C-9′ can influence the potency of the compounds. This finding can lead to the possibility of modifying the substituent at C-9′ of the oxyphenalenone scaffold and evaluating the antimicrobial and antibiofilm activities of their synthetic analogs.

## Figures and Tables

**Figure 1 marinedrugs-21-00194-f001:**
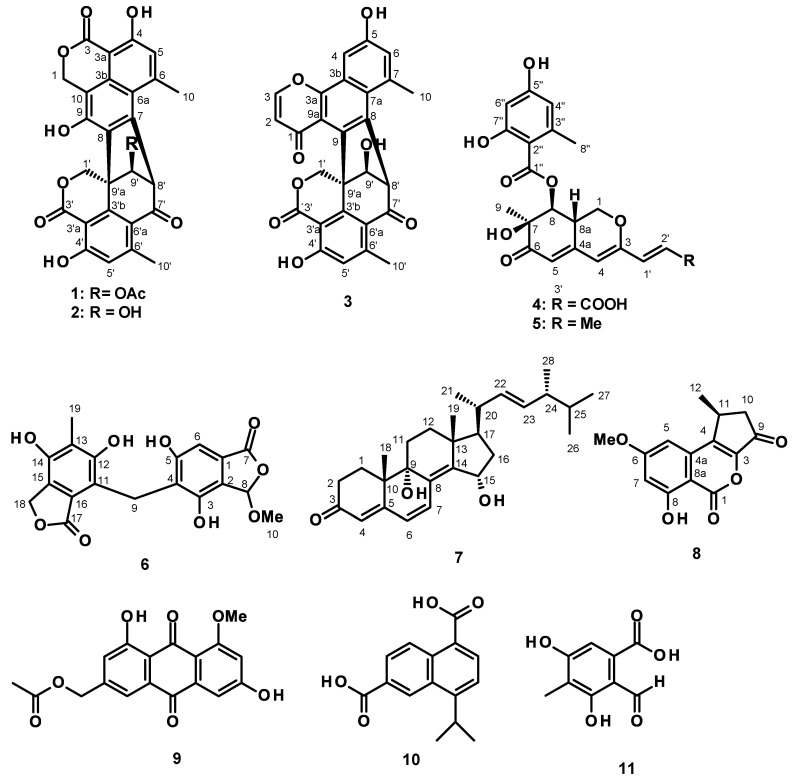
Structures of **1**–**11**.

**Figure 2 marinedrugs-21-00194-f002:**
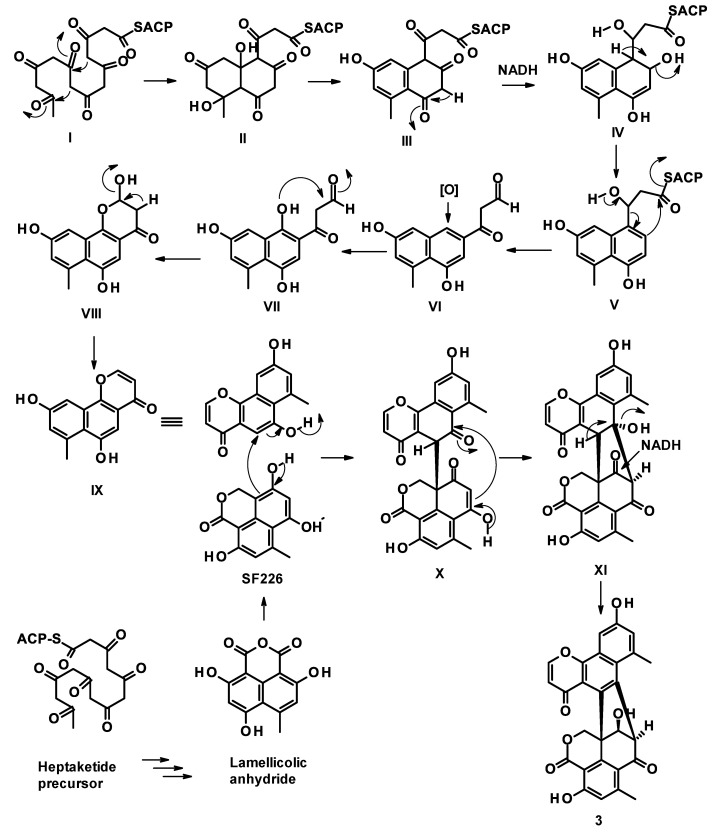
Proposed synthetic pathway for **3**.

**Figure 3 marinedrugs-21-00194-f003:**
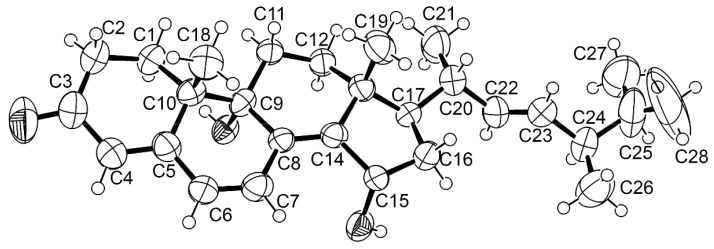
Ortep view of **7**.

**Figure 4 marinedrugs-21-00194-f004:**
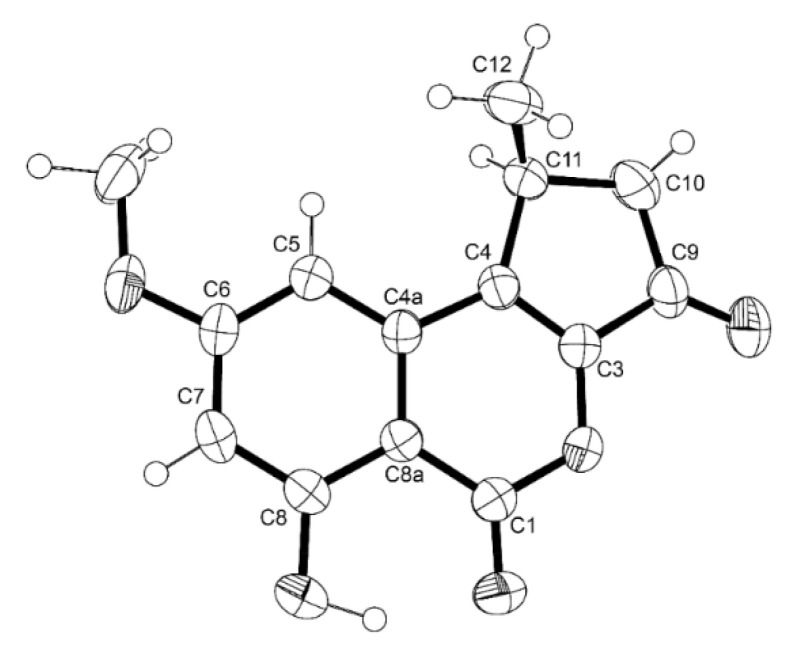
Ortep view of **8**.

**Table 1 marinedrugs-21-00194-t001:** ^1^H and ^13^C NMR data (DMSO-*d_6_*, 300 and 75 MHz), COSY, HMBC and ROESY for **2**.

Position	δ_C_, Type	δ_H_, (*J* in Hz)	COSY	HMBC	ROESY
1	67.2, CH_2_	5.64, d (15.1) 5.72, d (15.1)		C-3, 3b,9 C-3, 3b,9	
3	169.9, CO	-			
3a	97.9, C	-			
3b	137.3, C	-			
4	161.9, C	-			
5	119.5, CH	6.95, s	H-10	C-3, 4, 6a, 10	H_3_-10
6	146.4, C	-			
6a	119.5	-			
7	137.1, C	-			
8	135.2, C	-			
9	149.4, C	-			
9a	110.1, C	-			
10	24.8, CH_3_	2.97, s		C-5, 6	H-5
1′α β	70.5, CH_2_	4.98, d (12.4) 5.13 d (12.4)		C-3′, 3′b, 8, 9′, 9′b	
3′	168.3, CO				
3′a	104.2, C	-			
3′b	148.2, C	-			
4′	163.6, C	-			
5′	120.1, CH	6.82, d (0.6)	H-10′	C-3′a, 4′, 6′a, 10′	H_3_-10′
6′	152.8, C	-			
6′a	117,2, C	-			
7′	193.2, CO				
8′	65.0, CH	4.82,s		C-6′a, 7, 7′, 8, 9′, 9′a	H_3_-10
9′	85.7, CH	4.76, s		C-7, 8. 8′	H-1′β
9′a	50.0, C	-			
10′	23.7, CH_3_	2.48, s		C-5′, 6′ 6′a	H-5′
OH-4′		11.85, brs			
OH-9′		6.25, brs			

**Table 2 marinedrugs-21-00194-t002:** ^1^H and ^13^C NMR data (DMSO-*d_6_*, 300 and 75 MHz), COSY, HMBC and ROESY for **3**.

Position	δ_C_, Type	δ_H_, (*J* in Hz)	COSY	HMBC	ROESY
1	159.4, CO			C-3, 9a	
2	114.8, CH	6.46, d (10.0)	H-3	C-1,9a	H-3
3	141.2, CH	8.60, d (10.0)	H-2	C-1, 3a	H-2,4
3a	151.1, C	-			
3b	133.7, C	-			
4	103.9, CH	7.54, d (2.0	H-6	C-5, 6, 7a	H-3
5	157.3, C				
6	122.0, CH	7.04, d (2.0)-	H-4, H_3_-10	C-7a, 10a -	H_3_-10
7	137.9, C				
7a	122.4, C	-	-		
8	142.8, C	-			
9	130.3, C	-			
9a	112.6, C	-			
10	24.9, CH_3_	3.01, s	H-4, 6	C-6, 7, 7a	H-6, 8′
1′α β	70.1, CH_2_	5.04, d (12.6) 5.17, d (12.6)	1′β 1′α	C-3′, 3′b, 9, 9′a C-9	
3′	168.1, CO	-			
3′a	104.5, C	-			
3′b	148.1, C	-			
4′	161.0, C				
5′	121.0, CH	6.77, s	H_3_-10′	C-6′a	H_3_-10′
6′	152.0, C	-			
6′a	117.0, C				
7′	192.4, CO	-			
8′	65.7CH	4.97, s		C-7′, 8, 9, 9′, 9′a	H_3_-10
9′	86.0, CH	4.79		C-8, C-9	H-1′α
9′a	50.0, C	-			
10′	23.7, CH_3_	2.46, s	H-5′	C-5′, 6′, 6′a	H-6′
	OH-9′	6.33, br			
	OH-4′	10.26, br			

**Table 3 marinedrugs-21-00194-t003:** ^1^H and ^13^C NMR data (DMSO-*d6*, 300 and 75 MHz), COSY, HMBC and ROESY for **4**.

Position	δ_C_, Type	δ_H_, (*J* in Hz)	COSY	HMBC	ROESY
1α β	68.2, CH_2_	3.85, dd (11.0, 13.5) 4.71, dd (11.0, 5.2)	H-1β, 8a H-1α, 8a		H-8 H-1α, 8a
3	156.9, C	-			
4	110.5, CH	6.30, s		C-2′, 3, 5, 8a	H-5
4a	149.1, C	-			
5	119.7, CH	5.88, d (2.0)		C-4, 7, 8a	H-4
6	195.4, CO	-			
7	73.8, C	-			
8	75.2, CH	5.12, d (9.8)	H-8a	C-1, 1″, 8a	H_3_-1α, 9
8a	34.9, CH	3.25, m	H-1α, 1β, 8		
9	19.5, CH_3_	1.22, s		C-6, 7	H-8
1′	123.6, CH	6.22, d (15.5)		C-3, 3′	
2′	136.9, CH	7.05, d (15.5)	H-1′	C-3, 3′	
3′	167.3, CO	-			
1″	169.1, CO	-			
2″	109.9, C	-			
3″	161.0, C	-			
4″	100.7, CH	6.20, d (2.0)	H-6″	C-6″	
5″	159.5, C	-			
6″	109.9CH	6.19,brs	H-4″	C-4″, 8″	H_3_-8″
7″	140.0, C	-			
8″	21.6, CH_3_	2.30, s		C-2″, 6″, 7″	H-6″
OH-5″/7″	-	10.45, brs			
OH-7	-	5.82, br			

**Table 4 marinedrugs-21-00194-t004:** ^1^H and ^13^C NMR data (DMSO-*d6*, 300 and 75 MHz), COSY and HMBC for **6**.

Position	δ_C_, Type	δ_H_, (*J* in Hz)	COSY	HMBC
1	126.0, C	-		
2	126.2, C	-		
3	152.3, C	-		
4	121.4, C	-		
5	158.8, C	-		
6	101.4, CH	6.69, s	H-8, 9a, 9b	C-1, 4, 5, 7
7	169.0, CO	-		
8	102.8, CH	6.29, s	H_3_-9, 10	C-17, 10
9α β	18.3, CH_2_	4.35, d (14.4) 4.47, d (14.4)	H-6, 18 H-6, 18	C-3, 4, 5, 6, 11, 12 C-3, 4, 5, 6, 11, 12
10	56.6, OCH_3_	3.44, s	H-8	C-8
11	121.6, C	-		
12	155.2, C	-		
13	118.9, C	-		
14	148.2, C	-		
15	125.2, C	-		
16	117.0, C	-		
17	171.7, CO	-		
18	67.0, CH_2_	5.14, s	H-9, 19	C-11, 14, 15, 17
19	10.2, CH_3_	2.04, s	H-18	C-12, 13, 14
OH		9.42, s		

**Table 5 marinedrugs-21-00194-t005:** ^1^H and ^13^C NMR data (DMSO-*d6*_3_, 300 and 75 MHz), COSY and HMBC for **7**.

Position	δ_C_, Type	δ_H_, (*J* in Hz)	COSY	HMBC
1	27.5, CH_2_	1.66, m 1.81, m		
2	34.0, CH_2_	2.27, m 2.45, m		
3	198.6, CO	-		
4	124.9, CH	5.70, s		C-2, 6, 10
5	163.4, C	-		
6	124.8, CH	6.08, d (9.6)	H-7	C-4, 5, 8, 10
7	132.6, CH	7.04, d (9.6)	H-6	C-5, 8, 9, 14
8	131.3, C	-		
9	71.8, C	-		
10	42.0, C	-		
11	26.4, CH_2_	1.54, m 1.95, m		
12	31.7, CH_2_	1.66, m 1.81, m		
13	44.5, C	-		
14	158.2, C	-		
15	68.4, CH	4.64, m	H-16	
16	40.3, CH_2_	1.69, m	H-15	
17	52.5, CH	1.60, m		
18	21.0, CH_3_	1.04, s		C-1, 5, 9, 10
19	19.3, CH_3_	0.85, s		C-13, 14, 17
20	38.5, CH	2.08, m	H-21, 22	
21	21.7, CH_3_	1.05, d (6.4)	H-20	C-17, 22
22	135.5, CH	5.25, m	H-20	C-24
23	132.3, CH	5.24, m	H-24	C-20
24	42.6, CH	1.87, m	H-23, 26	
25	33.0, CH	1.47, m	H-28	
26	17.9, CH_3_	0.92, d (6.8)	H-24	C-23, 24, 25
27	19.9, CH_3_	0.84, d (6.8)		C-24, 25, 28
28	20.3, CH_3_	0.82, d (6.8)	H-25	C-24, 25, 27
OH-9	-	4.33, s		C-8, 9, 10

**Table 6 marinedrugs-21-00194-t006:** Antibacterial activity of compounds **1**, **2**, **4**–**8**, **9** and **10** against Gram-positive reference and MRSA *S. aureus* 74/24. MIC and MBC are expressed in µg/mL.

Compound	*E. faecalis* ATCC 29212		*S. aureus* ATCC 29213		*S. aureus* 74/24 (MRSA)	
	MIC	MBC	MIC	MBC	MIC	MBC
**1**	>64	>64	4	>64	4	>64
**2**	32	>64	8	>64	16	>64
**4**	>64	>64	>64	>64	>64	>64
**5**	>64	>64	64	>64	16	>64
**6**	>64	>64	>64	>64	>64	>64
**7**	>64	>64	>64	>64	>64	>64
**8**	>64	>32	>32	>32	>32	>32
**10**	>64	>64	>64	>64	>64	>64
**11**	>32	>32	>32	>32	>32	>32
VAN	4	-	-	-	-	-
OXA	-	-	0.2	-	64	-

MIC, minimum inhibitory concentration; MBC, minimum bactericidal concentration; VAN, vancomycin; OXA, oxacillin.

**Table 7 marinedrugs-21-00194-t007:** Percentage of biofilm formation for compounds that showed antibiofilm activity after 24 h incubation.

Compound	Concentration	Biofilm Biomass (% of Control)	
		*E. faecalis* ATCC 29212	*S. aureus* ATCC 29213
**1**	8 µg/mL	-	0.08 ± 0.03 *** (2xMIC)
	4 µg/mL	-	0.19 ± 0.17 *** (MIC)
**2**	64 µg/mL	0.13 ± 0.02 *** (2xMIC)	-
	16 µg/mL	-	0.13 ± 0.05 *** (2xMIC)
	8 µg/mL	-	0.29 ± 0.13 *** (MIC)
DMSO	1% (*v*/*v*)	0.99 ± 0.02 ***	1.00 ± 0.02 ***

Data are shown as mean ± SD of at least three independent assays. MIC, minimal inhibitory concentration. One-sample *t*-test: *** *p* < 0.001 significantly different from 100%.

## Data Availability

The original data presented in the study are included in the article/Appendix A; further inquiries can be directed to the corresponding authors.

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
