# Peer review of "New Hybrid Phenalenone Dimer, Highly Conjugated Dihydroxylated C28 Steroid and Azaphilone from the Culture Extract of a Marine Sponge-Associated Fungus, Talaromyces pinophilus KUFA 1767"

_marinedrugs, 2023, doi:10.3390/md21030194_

Round 1

Reviewer 1 Report

The genus Talaromyces attracted considerable attention from natural product chemists because it contains reservoirs of chemo- and bio-biodiversity. This work exemplified the potentials of producing diverse new compounds from the genus. The structure elucidation of this work seems sound and corrected supported by detailed NMR and X-ray crystallographic analysis. The manuscript is well written, and can be considered for publication after some minor revision.

1.      (M+H) should be [M+H].

2.      Table 4. DMSO-d63?

3.      COSY should be 1H 1H COSY.

4.      Three-line table is recommended.

5.      A 2D NMR correlation Figure of new compounds is recommended.

Author Response

Reviewer #1

The genus Talaromyces attracted considerable attention from natural product chemists because it contains reservoirs of chemo- and bio-biodiversity. This work exemplified the potentials of producing diverse new compounds from the genus. The structure elucidation of this work seems sound and corrected supported by detailed NMR and X-ray crystallographic analysis. The manuscript is well written, and can be considered for publication after some minor revision.

Reply: We wish to thank reviewer #1 for his/her appreciation of our manuscript.

Reviewer #1

(M+H) should be [M+H].

Reply: the parenthesis of (M+H)+ was corrected to [M+H]+ throughout the manuscript.

Reviewer #1

Table 4. DMSO-d63?

Reply: The typo of Table 4 was corrected.

Reviewer #1

COSY should be 11H COSY.

Reply: The term COSY is widely accepted for the 2D proton spectrum. Although some aurhors use 1H-1H COSY as reviewer #1 suggested, most of the authors use only COSY. We have adopted the term “COSY” for all of our published papers.

Reviewer #1

Three-line table is recommended.

Reply: We don’t understand what reviewer#1 means by “three-line table”. All the tables are elaborated in the same format as those in the aricles we have published in Marine Drugs recently.

Reviewer #1

A 2D NMR correlation Figure of new compounds is recommended.

Reply. All the 2D NMR correlations are already in the tables of all the new compounds. The figures of 2D NMR correlations will not facilitate the understanding of the correlations.More often than not, the arrows used to indicate HMBC and NOESY correlations are so dense that they rather make more confusion than clarification. Moreover, it does not make sense to have the 2D NMR correlations in the tables and repeated them in the figures. The key 2D NMR correlations should be used when the tables do not indicate these correlations.

Reviewer 2 Report

The manuscript reports the isolation and structure elucidation of four structurally diverse new compounds 3-4 and 6-7 from the culture extract of the marine sponge-associated fungus Talaromyces pinophilus KUFA 1767, together with seven known compounds 1-2, 5, and 8-11. The structure elucidation of the new compounds was solved through examination of their mass and their 1D and 2D NMR data. The absolute configuration at C-9’ for the two previously reported oxyphenalenone dimers 1 and 2 was revised and confirmed by ROESY correlations for compound 2. A plausible biosynthetic pathway for compound 3 is presented. The absolute configuration of compounds 7 and 8 was investigated by X-ray crystallographic analysis. Evaluation of the antibacterial activity against two Gram-positive and two Gram-negative strains as well as against multidrug-resistant strains is presented. Compounds 1 and 2 showed significant antibacterial activity against both S. aureus and MRSA strains, as well as their ability to inhibit a biofilm formation in S. aureus ATTC 29213. These results suggest that the dimeric oxyphenalenone scaffold is essential for the antibacterial and antibiofilm activities as well as the nature of the substituents at C-9’.

The manuscript is methodologically sound and well written. The results are clearly presented.

Some questions that arose during the reading of the manuscript are listed below:

- The authors claim that the study was conducted in an ongoing search for antibiotic and antibiofilm compounds. 

This is a very relevant objective, as new antibiotics are urgently needed due to the emergence of multidrug-resistant bacterial strains. However, this should be developed further in the introduction. This exciting research combining the search for antibacterial and antibiofilm compounds deserves more detail. 

- Why was the fungus T. pinophiluschosen for this study? Did its crude extract show potent antibacterial activity? As the authors did not perform bio-guided fractionation, additional information on the procedure used should be provided.

Typographical errors are detailed in the attached manuscript and some revised sentences are suggested.

Author Response

Reviewer#2

The manuscript is methodologically sound and well written. The results are clearly presented.

 Reply: We wish to thank reviewer #2 for his/her appreciation of our manuscript.

Some questions that arose during the reading of the manuscript are listed below:

Reviewer#2

- The authors claim that the study was conducted in an ongoing search for antibiotic and antibiofilm compounds. 

This is a very relevant objective, as new antibiotics are urgently needed due to the emergence of multidrug-resistant bacterial strains. However, this should be developed further in the introduction. This exciting research combining the search for antibacterial and antibiofilm compounds deserves more detail. 

Reply: We thank reviewer #2 for his or her suggestion to add the concept of emergence of multidrug-resistant bacteria and the importance of the biofilm in the virulence of these pathogenic bacteria. However, we will maintain the introduction as it is as we have been working on this subject for around 10 years and we have written this concept in the introduction in our previous papers such as in reference 34 (Gomes, N. M.; Bessa, L. J.; Buttachon, S.; Costa, P. M.; Buaruang, J.; Dethoup, T.; Silva, A. M. S.; Kijjoa, A. Antibacterial and antibiofilm activities of tryptoquivalines and meroditerpenes isolated from the marine-derived fungi Neosartorya paulistensis, N. laciniosa, N. tsunodae, and the soil fungi N. fischeri and N. siamensis. Mar. Drugs 2014, 12, 822-839. https://doi.org/10.3390/md12020822). Therefore, it does not make sense to repeat the same thing that we had written in our previous papers.

As reviewer #2 can see from our papers that we have published on this subject (below) that we did not repeat this concept that we have written in Ref. 34 which is our first article on this subject.

  1. Nelson M. Gomes, Lucinda J. Bessa, Suradet Buttachon, Paulo M. costa, Jamrearn Buaruang, Tida Dethoup, Artur. M. S. Silva and Anake Kijjoa (2014). Antibacterial and Antibiofilm Activity of Tryptoquivalines and Meroditerpenes from Marine-Derived Fungi Neosartorya paulistensis, laciniosa, N. tsunodae, and the soil fungi N. fischeri and N. siamensis. Mar. Drugs 12, 822-839. https://doi.org/10.3390/md12020822.
  2. Chadaporn Prompanya, Tida Dethoup, Lucinda J. Bessa, Madalena M. M. Pinto, Luís Gales, Paulo M. Costa, Artur M. S. Silva, Anake Kijjoa (2014). New Isocoumarin Derivatives and Meroterpenoids from the Marine Sponge-Associated Fungus Aspergillus similanensis nov. KUFA 0013. Mar. Drugs 12, 5160-5173. https://doi.org/10.3390/md12105160.
  3. Chadaporn Prompanya, Carla Fernandes, Sara Cravo, Madalena M. M. Pinto, Tida Dethoup, Artur M. S. Silva, Anake Kijjoa (2015). A New Cyclic Hexapeptide and a New Isocoumarin Derivative from the Marine Sponge-Associated Fungus Aspergillus similanensis KUFA 0013. Drugs 13, 1432-1450. https://doi.org/10.3390/md13031432.
  4. War War May Zin, Suradet Buttachon, Jamrearn Buaruang, Luís Gales,
    José A. Pereira, Madalena M.M. Pinto, Artur M.S. Silva and Anake Kijjoa (2015). A New Meroditerpene and a New Tryptoquivaline from the Algicolous Fungus Neosartorya takakii KUFC 7898. Drugs 13, 3776-3790. https://doi.org/10.3390/md13063776.
  5. Suradet Buttachon, War War May Zin, Tida Dethoup, Luís Gales, José A. Pereira, Artur M. S. Silva,Anake Kijjoa (2016). Secondary Metabolites from the Culture of the Marine Sponge-Associated Fungi Talaromyces tratensis and Sporidesmium circinophorum. Planta Med. 82, 888-896. doi: 1055/s-0042-103687.
  6. Chadaporn Prompanya, Tida Dethoup, Luís Gales, Michael Lee, José A. C. Pereira,Artur M. S. Silva, Madalena M. M. Pinto and Anake Kijjoa (2016). New polyketides and new benzoic acid derivatives from the marine sponge-associated fungus Neosartorya quadricincta KUFA 0081. Mar Drugs 14, 134. https://doi.org/10.3390/md14070134.
  7. War War, Suradet Buttachon, Tida Dethoup, Carla Fernandes, Sara Cravo, Madalena M. M. Pinto, Luís Gales, José A. Pereira, Artur M. S. Silva, Nazim Sekeroglu and Anake Kijjoa (2016). New Cyclotetrapeptides and a New Diketopiperzine Derivative from the Marine Sponge-Associated Fungus Neosartorya glabra KUFA 0702. Mar Drugs 14, 136.https://doi.org/10.3390/md14070136.
  8. Lucinda J. Bessa, Suradet Buttachon, Tida Dethoup, Rosário Martins, Vitor Vasconcelos, Anake Kijjoa and Paulo Martins da Costa (2016). Neofiscalin A and fiscalin C are potential novel indole alkaloid alternatives for the treatment of multidrug resistant Gram-positive bacterial infections. FEMS Microbiol- Lett. 363, fnw150. https://doi.org/10.1093/femsle/fnw150.
  9. War War May Zin, Suradet Buttachon, Tida Dethoup, José A. Pereira, Luís Gales, Ângela Inácio, Paulo M. Costa, Michael Lee, Nazim Sekeroglu, Artur M. S. Silva, Madalena M. M. Pinto, Anake Kijjoa (2017). Antibacterial and antibiofilm activities of the metabolites isolated from the culture of the mangrove-derived endophytic fungus Eurotium chevalieri Phytochemistry 141, 86-97.https://doi.org/10.1016/j.phytochem.2017.05.015.
  10. Decha Kumla, Tin Shine Aung, Suradet Buttachon, Tida Dethoup, Luís Gales, José A. Pereira, Ângela Inácio, Paulo M. Costa, Michael Lee, Nazim Sekeroglu, Artur M. S. Silva, Madalena M. M. Pinto, Anake Kijjoa (2017). A New Dihydrochromone Dimer and Other Secondary Metabolites from Cultures of the Marine Sponge-Associated Fungi Neosartorya fennelliae KUFA 0811 and Neosartorya tsunodae KUFC 9213. Drugs 15, 375; doi:10.3390/md15120375.
  11. Suradet Buttachon, Alice A. Ramos, Ângela Inácio, Tida Dethoup, Luís Gales, Michael Lee, Paulo M. Costa, Artur M. S. Silva, Nazim Sekeroglu, Eduardo Rocha, Madalena M. M. Pinto, José A. Pereira, Anake Kijjoa (2018). Bis-Indolyl Benzenoids, Hydroxypyrrolidine Derivatives and Other Constituents from Cultures of the Marine Sponge-Associated Fungus Aspergillus candidus Mar. Drugs16, 119; doi:10.3390/md16040119.
  12. Decha Kumla, José A. Pereira, Tida Dethoup, Luis Gales, Joana Freitas-Silva, Paulo M. Costa, Michael Lee, Artur M. S. Silva, Nazim Sekeroglu, Madalena M. M. Pinto and Anake Kijjoa (2018). Chromone Derivatives and Other Constituents from Cultures of the Marine Sponge-Associated Fungus Penicillium erubescens KUFA0220 and Their Antibacterial Activity. Drugs 16, 289; doi:10.3390/md16080289.
  13. Decha Kumla, Tida Dethoup, Luis Gales, José A. Pereira, Joana Freitas-Silva, Paulo M. Costa, Artur M. S. Silva, Madalena M. M. Pinto and Anake Kijjoa (2019). Erubescensoic acid, a new polyketide and a xanthonopyrone SPF-3059-26 from the culture of the marine sponge-associated fungus Penicillium erubescens KUFA 0220 and Antibacterial Activity Evaluation of some of its constitutuents. Molecules 24, Doi: 10.3390/molecules24010208.
  14. Solida Long, Diana I. S. P. Resende, Andreia Palmeira, Anake Kijjoa, Artur M. S. Silva, Maria Elizabeth Tiritan, Patr´ıcia Pereira-Terra, Joana Freitas-Silva, Sandra Barreiro, Renata Silva, f Fernando Remiao, Eugenia Pinto, Paulo Martins da Costa, Emília Sousa, Madalena M. M. Pinto (2020). New marine-derived indolymethyl pyrazinoquinazoline alkaloids with promising antimicrobial profiles. RSC Adv. 10, 31187-31204. doi: 10.1039/d0ra05319h.
  15. Decha Kumla, Emilia Sousa, Alessia Marengo, Tida Dethoup, José A. Pereira, Luís Gales, Joana Freitas-Silva, Paulo M. Costa, Sharad Mistry, Artur M.S. Silva, Anake Kijjoa (2021). 1,3-Dioxepine and spiropyran derivatives of viomellein and other dimeric naphthopyranones from cultures of Aspergillus elegans KUFA0015 and their antibacterial activity. Phytochemistry 181, 112575. https://doi.org/10.1016/j.phytochem.2020.112575.
  16. Fatima P. Machado, Decha Kumla, José A. Pereira, Emilia Sousa, Tida Dethoup, Joana Freitas-Silva, Paulo M. Costa, Sharad Mistry, Artur M.S. Silva, Anake Kijjoa (2021). Prenylated phenylbutyrolactones from cultures of a marine sponge-associated fungus Aspergillus flavipes Phytochemistry 185, 112709. https://doi.org/10.1016/j.phytochem.2021.112709.
  17. Joana D. M. de Sá, José A. Pereira, Tida Dethoup, Honorina Cidade, Maria Emília Sousa, Inês C. Rodrigues, Paulo M. Costa, Sharad Mistry, Artur M. S. Silva, Anake Kijjoa (2021). Anthraquinones, Diphenyl Ethers, and Their Derivatives from the Culture of the Marine Sponge-Associated Fungus Neosartorya spinosa KUFA 1047. Mar Drugs 19, 457. https://doi.org/10.3390/md19080457.
  18. Fernando Durães, Nikoletta Szemerédi, Decha Kumla, Madalena Pinto, Anake Kijjoa, Gabriella Spengler, Emília Sousa (2021). Metabolites fromMarine-Derived Fungi as Potential Antimicrobial Adjuvants. Drugs 19, 475; https://doi.org/10.3390/md19090475.
  19. Fátima P. Machado, Inês C. Rodrigues, Luís Gales, José A. Pereira, Paulo M. Costa, Tida Dethoup, Sharad Mistry, Artur M. S. Silva, Vitor Vasconcelos and Anake Kijjoa (2022). New alkylpyridinium anthraquinone, isocoumarin, C-glucosyl resorcinol derivative and prenylated
    pyranoxanthones from the culture of a marine sponge-associated fungus, Aspergillus stellatus KUFA 2017. Drugs 20, 672;https://doi.org/10.3390/md20110672.

Reviewer#2

- Why was the fungus T. pinophilus chosen for this study? Did its crude extract show potent antibacterial activity? As the authors did not perform bio-guided fractionation, additional information on the procedure used should be provided.

Reply:

First, the reasons why we chose the strain KUFA 1767 of T. pinophillus is well described in the introduction. Secondly, we are not in favour of using bio-guided fractionation because by using this method we would discard many interesting compounds that do not have the bioactivity that we use to monitor the fractionation. We use the isolated compound approach for our research. That means we are focusing on chemical diversity among different strains of the fungi that are isolated from different sources and environments. This research is part of the project of PT-Open Screen. That means besides antibacterial and antibiofilm activities, the isolated compounds will be screened for other activities by other groups in our research center too. However, our group is focusing on finding new compounds with antibacterial and antibiofilm activities. That is why we assayed all the isolated compounds (both undescribed and previously reported) for the reference and multidrug-resistant bacterial strains and evaluated their capacity to inhibit biofilm formation first.

Reviewer#2

Typographical errors are detailed in the attached manuscript and some revised sentences are suggested.

Reply: We would like to thank reviewer #2 for checking all the typos as well as making suggestion to modify some sentences. We have made corrections of the typos that reviewer #2 suggested. However, we have maintained the sentences that we think they are correctly written in clear English.